# A performance comparison of variable encoding techniques for QUIO and QUBO problems

Pedro Maciel Xavier[1][], Yirang Park[1][0009−0008−6629−3308], Diego Noble[2][], Italo Santana[2][], Miguel Paredes[2][], and David E. Bernal Neira[1][0000−0002−8308−5016]

[1] Davidson School of Chemical Engineering, Purdue University, 480 Stadium Mall Drive, West Lafayette, IN 47907, USA
dbernaln@purdue.edu
[2] Dell Technologies

**Abstract.** In this paper, we explore the encoding Quadratic Integer Optimization (QIP) problems into Quadratic Unconstrained Integer Optimization (QUIO) formulations with a target integer basis. These formulations are suitable for solving on quantum computers based on qudits, which natively extend the integer representation of their qubit-based counterparts. For qubit-based computers, the usual framework for representing discrete optimization problems is the Quadratic Unconstrained Binary Optimization (QUBO) formulation. Decision variables can, for QUBOs, assume only binary values, while for QUIOs, they can represent integer values from zero up to a machine-dependent maximum. One advantage of working with a larger domain of decision variables is that it enables QIP problems to be cast as QUIO formulations. As our results highlight, these formulations use fewer decision variables than, for example, Quadratic Unconstrained Binary Optimization (QUBO)-based formulations. We selected various candidate problems to empirically verify our reformulations: Quadratic Facility Location Problem, Quadratic Inventory Management Problem, Quadratic Vehicle Routing Problem, and Quadratic Knapsack Problem. Problem instances for these problems are diverse in data distribution and size and are reformulated into QUBO and QUIO. We compare these formulations, characterizing them using metrics associated with solution performance. Moreover, using qubit- and qudit-based entropy quantum machines, we compared the performance of the resulting formulations for instances amenable to these architectures. Our primary goal is to conduct computational experiments to verify the impacts of each encoding type on these problems, aiming to find insights that could potentially generalize to similar optimization problems. Moreover, we also indicate guidelines to accelerate the encoding process, ensuring that the potential quantum advantage using qudit quantum computers is not lost during the classical pre-processing and during problem reformulation. Finally, we provide open-source software to perform the reformulations and communicate them with qudit-based entropy quantum devices, allowing others to map and solve QIP problems using QUIO reformulations.

**Keywords:** QIP · QUIO · QUBO · variable encoding comparison

## 1 Introduction

Quadratic Integer Problems (QIPs) belong to a class of computational problems in which the objective is a quadratic function and the constraints are linear. Problems of this class can be found in many fields, including engineering, finance, and logistics[12]. Although immensely versatile for modeling, QIP is in the worst-case complexity class of NP [9]. Beyond traditional solvers, such

as branch-and-bound algorithms or heuristics executed on classical computers, QIPs can also be addressed by quantum computers via reformulating the problem into a Quadratic Unconstrained Integer Optimization (QUIO) problem or into a Quadratic Unconstrained Binary Optimization (QUBO) problem. Quadratic nonlinearity in the objective aligns well with native operations in many quantum algorithms, making QIPs particularly amenable to quantum formulations.

Mapping constrained optimization problems into formulations amenable to quantum computation is essential for leveraging emerging (quantum) hardware. This mapping is not unique and might reveal strengths associated with a given quantum solver, strengths that often diverge from classical optimization heuristics. Although progress has been made in mapping constrained problems, such as Quadratic Unconstrained Binary Optimization (QUBO) methods [14], traditional quantum architectures and algorithms are not inherently tied to the QUBO formalism.

In general terms, a QUIO problem encodes decision variables using an integer basis, whereas a QUBO problem encodes the same variables using a binary basis, thus making QUIO a more general form of QUBO. For both, the resulting formulation is derived from a QIP problem by introducing slack variables $s \in \mathbb{Z}^m$ for QUIO and $s \in \{0,1\}^m$ for QUBO, and penalizing infeasibility in the objective function instead of enforcing $Ax \leq b$ directly.

Recently, a class of quantum computers capable of encoding information in an integer basis has emerged. These machines are based on qudits, which are a generalization of qubits that allow for a dimensional extension of the computational basis of qubits. In other words, qudits can represent more than two states, contrary to (0 and 1), as natively represented using qubits. From the perspective where a QIP problem has decision variables with a broad domain larger than binary, a reformulation of a QIP into QUIO can be more suitable, as the QUIO can be solved directly on these new devices rather than encoding the integer variables into a binary (e.g., one-hot encoding), as seen in QUBO form. It is essential to highlight that a qudit is not range-free due to physical and measurement limitations. Additionally, problems formulated as QUBOs can be directly solved on qubit-based quantum systems (e.g., quantum annealers), which are more mature technologies than qudit-based devices and typically have a larger number of qubits. This trade-off, therefore, implies that a QIP recast as QUBO may be solved better than a QUIO, even considering the required number of qubits to represent this problem. This, of course, needs to account for all forms of pre- and post-processing tasks (e.g., problem compilation, solution decoding, etc.) when considering either a QUIO or QUBO formulation.

In this study, our objective was to investigate the different encoding types for QUIO and QUBO and their impact on space utilization before solving and performing. In detail, we highlight the following contributions that are covered in this paper:

1. Define four classes of QIPs and describe the procedure for generating their problem instances.
2. Conduct computational experiments over these problem instances using variable encoding techniques for QUBOs and their comparison with QUIOs.
3. Assess how different encoding techniques influence the performance of the QUIO and QUBO formulations.

The remainder of this paper is structured as follows. Section 2 provides an overview of the literature on problem reformulations foc QUBOs and QUIOs. In Section 3, we review the mathematical forms of QIPs and their formulations into QUIOs and QUBOs. Section 4 explains the implementation details of the QUBO and QUIO reformulations. Next, Section 5 details the optimization problems and their corresponding formulations. Section 6 reports the experimental results on the tested encoding strategies, both on the resulting reformulated problem and when running them on

a qudit-based Entropy Quantum Computer. Finally, Section 7 presents final remarks and outlines research gaps that could be addressed in future work.

## 2   Literature Review

Quantum optimization has largely been centered around *Quadratic Unconstrained Binary Optimization (QUBO)* formulations, which map discrete optimization problems into forms compatible with the transverse field Ising spin model [5]. This mapping is well aligned with hardware based on qubits, where variables assume binary values, and has been widely used in quantum gate-based and annealing systems.

To reformulate more general discrete problems, such as *Quadratic Integer Programs (QIPs)* into QUBOs, classical pre-processing is required. Typical steps include binarizing integer variables, introducing slack variables to encode inequalities, and penalizing constraint violations in the objective function [14]. These steps often lead to significant growth in problem size, increased density of quadratic terms, and degradation in sparsity. Such transformations can impose burdens on quantum devices, particularly when embedding and calibration budgets are limited.

Recently, attention has shifted to quantum hardware built from *qudits*, or *d*-level quantum systems, which generalize the qubit model [13]. Qudits can represent larger alphabets natively, reducing the need for binary encodings and enabling more compact problem representations. In gate-based settings, qudit circuits have been shown to reduce circuit depth and offer improved connectivity and expressivity [1, 13]. Such hardware capabilities enable a more natural and efficient representation of integer variables, motivating the development of integer-native reformulations.

The idea of exploiting qudit capabilities has been explored in algorithm design for certain structured problems [11, 2]. A broader overview of high-dimensional quantum computing also highlights scenarios where qudits offer both algorithmic and architectural benefits [13].

However, these efforts have not extended to general-purpose reformulations of QIPs into *Quadratic Unconstrained Integer Optimization (QUIO)* problems. To our knowledge, there are currently no systematic studies or software infrastructure for the direct reformulation of QIPs to QUIOs designed for qudit-compatible quantum optimization. This is a notable gap, especially given the growing availability of early-stage qudit hardware.

Libraries such as QUBO.jl [14] extensively support QUBO reformulations, but did not use to provide analogous support for QUIO targets. This work aims to fill that gap by introducing structured reformulations for QUIO, comparing them to standard QUBO approaches, and empirically evaluating them across representative QIP classes. By leveraging direct integer encodings, QUIO reformulations promise to improve scalability and maintain sparsity, aligning well with current and future qudit-based computing capabilities.

## 3   Mathematical formulations for QIPs, QUIOs, and QUBOs

In this section, we reformulate optimization problems with quadratic objectives, linear constraints, and integer variables into quadratic integer programs (QIPs). We then convert these QIPs into quadratic unconstrained integer optimization (QUIO) problems. QUIOs extend QUBOs (Quadratic Unconstrained Binary Optimization), which require binary encodings and a significantly larger number of variables.

QUIOs directly exploit quantum hardware with *qudits* (multi-level systems), unlike QUBOs, which rely on *qubits* (two-level systems). The availability of prototype qudit-based machines motivates the use of direct integer formulations, as QUIOs preserve sparsity and circumvent the exponential blow-up of binary encodings in QUBOs.

We consider problems of the form:

$$\min_{x \in \{0,\dots,U\}^n} x^\top Q x + c^\top x \quad \text{s.t.} \quad Ax \leq b, \tag{1}$$

with $Q \in \mathbb{R}^{n \times n}$, $A \in \mathbb{R}^{m \times n}$, $b \in \mathbb{R}^m$. Since QIPs are discrete, we assume without loss of generality that their data can be scaled to integer values: $Q \in \mathbb{Z}^{n \times n}$, $A \in \mathbb{Z}^{m \times n}$, $b \in \mathbb{Z}^m$. For all notations used in this section, we refer the reader to Table 1. Therefore, we consider QIPs in the following form:

$$\min_{x \in \mathbb{Z}^n} x^\top Q x + c^\top x \quad \text{s.t.} \quad Ax \leq b, \qquad Q \in \mathbb{Z}^{n \times n}, \ A \in \mathbb{Z}^{m \times n}, \ b \in \mathbb{Z}^m. \tag{2}$$

Table 1: Notation summary for QIP, and QUIO and QUBO reformulations.

| Symbol | Description |
|---|---|
| $n$ | Number of original decision variables |
| $m$ | Number of linear inequality constraints |
| $U$ | Upper bound on integer variables $x_i \in \{0, \dots, U_i\}$ |
| $Q \in \mathbb{Z}^{n \times n}$ | Quadratic coefficient matrix |
| $c \in \mathbb{Z}^n$ | Linear coefficient vector |
| $A \in \mathbb{Z}^{m \times n}$ | Constraint coefficient matrix |
| $b \in \mathbb{Z}^m$ | Constraint right-hand side |
| $x \in \mathbb{Z}^n$ | Original integer decision variables |
| $s \in \mathbb{Z}^m$ | Integer slack variables for relaxed constraints |
| $z = [x; s] \in \mathbb{Z}^{n+m}$ | QUIO decision vector |
| $D = \text{diag}(\mu)$ | Penalty weight matrix for QUIO constraints |
| $G \in \mathbb{Z}^{(n+m) \times (n+m)}$ | Block-quadratic matrix in QUIO form |
| $N$ | Number of binary variables encoding $x$ in QUBO |
| $K$ | Number of binary slack variables in QUBO |
| $E \in \mathbb{Z}^{n \times N}$ | Binary encoding matrix for variables |
| $T \in \mathbb{Z}^{m \times K}$ | Binary slack encoding matrix |
| $z \in \{0, 1\}^N$ | Binary vector encoding original variables in QUBO |
| $t \in \{0, 1\}^K$ | Binary vector encoding slack variables in QUBO |
| $w = [z; t] \in \{0, 1\}^{N+K}$ | Full QUBO decision vector |
| $\Lambda = \text{diag}(\lambda)$ | Penalty weight matrix in QUBO |
| $H \in \mathbb{Z}^{(N+K) \times (N+K)}$ | QUBO quadratic cost matrix |

We demonstrate, in Table 2, a summary of the main differences among the QIP, QUBO, and QUIO. In Subsections 3.1 and 3.2, we present the formal details of QIPs reformulated to QUIO and QUBO.

Table 2: Comparison of QIP, QUBO, and QUIO formulations and their connection to quantum hardware.

| Acronym | Decision variables | Constraints | Link to quantum hardware |
|---------|--------------------|-------------|--------------------------|
| QIP  | integers $x \in \mathbb{Z}^n$ | linear $Ax \leq b$ | natural mixed-integer model |
| QUBO | binaries $z \in \{0,1\}^N$ | none | matches qubits (2-level) |
| QUIO | integers $w \in \mathbb{Z}^\ell$ | none | matches *qudits* (multi-level) |

## 3.1 QUIO Reformulation (with Slack)

A reformulation of QIP (Equation 2) to QUIO requires introducing slack variables $s \in \mathbb{Z}^m$ and penalize infeasibility in the objective function instead of enforcing $Ax \leq b$ directly. We define residual as $r := Ax + s - b$, and weight violations using $\mu \in \mathbb{R}_+^m$, with $M = \mathrm{diag}(\mu)$, resulting in the following form:

$$\min_{x \in \mathbb{Z}^n, \, s \in \mathbb{Z}^m} x^\top Q x + c^\top x + r^\top M r, \quad \text{where } r := Ax + s - b. \tag{3}$$

**Block-Quadratic Form in $z = [x; s]$** Define:

$$z = \begin{bmatrix} x \\ s \end{bmatrix} \in \mathbb{Z}^{n+m}, \quad r := Ax + s - b.$$

Expanding $r^\top M r$ yields:

$$\boxed{\min_{z \in \mathbb{Z}^{n+m}} z^\top G z + g^\top z + \text{const}} \tag{4}$$

where

$$G = \begin{bmatrix} Q + A^\top M A & A^\top M \\ MA & M \end{bmatrix}, \quad g = \begin{bmatrix} c - 2A^\top M b \\ -2Mb \end{bmatrix}, \quad \text{const} = b^\top M b.$$

This reformulation removes constraints from the QIP and yields a QUIO compatible with qudit hardware. In the next section, we compare this to the traditional QUBO encoding.

## 3.2 QUBO Reformulation for Comparison

**Binary Encoding of $x$** To encode $x_i \in \{0, \ldots, U_i\}$, represent each component using $\lceil \log_2 U_i \rceil$ binary variables. Let:

$$x = Ez, \quad z \in \{0,1\}^N, \quad N \sim \sum_i \lceil \log_2 U_i \rceil,$$

where $E \in \mathbb{Z}^{n \times N}$ is the binary encoding matrix. This quantity represents a lower bound on the number of binary variables needed to encode $x$.

**Constraint Encoding** Transform $Ax \leq b$ into:

$$Bz \leq d, \quad B := AE, \quad d := b.$$

**Slack and Penalty** Introduce binary slack $t \in \{0,1\}^K$ such that:

$$Bz + Tt = d, \quad T \in \mathbb{Z}^{m \times K}, \quad K \sim m \lceil \log_2 V \rceil$$

for some slack bound $V$. Use penalty weights $\lambda \in \mathbb{R}_+^m$.

$$\min_{z \in \{0,1\}^N, \, t \in \{0,1\}^K} z^\top E^\top Q E z + (Bz + Tt - d)^\top \operatorname{diag}(\lambda)(Bz + Tt - d). \tag{5}$$

Here, $E^\top Q E$ is interpreted as the lifted quadratic term acting on the binary expansion $z$ of the original variables. Constraint violation is then penalized via binary slack encoding. Notice that one needs to choose a large enough penalty factor to enforce the constraints.

**Quadratic Form in $w = [z; t]$** Define $w = \begin{bmatrix} z \\ t \end{bmatrix} \in \{0,1\}^{N+K}$, where $z \in \{0,1\}^N$ encodes the original integer variables and $t \in \{0,1\}^K$ encodes the binary slack variables. The expanded form becomes:

$$\boxed{\min_{w \in \{0,1\}^{N+K}} w^\top H w + h^\top w + \text{const}} \tag{6}$$

where the matrix $H \in \mathbb{Z}^{(N+K) \times (N+K)}$ and vector $h \in \mathbb{Z}^{N+K}$ are defined as:

$$H = \begin{bmatrix} E^\top Q E + B^\top \Lambda B & B^\top \Lambda T \\ T^\top \Lambda B & T^\top \Lambda T \end{bmatrix}, \quad h = \begin{bmatrix} E^\top c - 2B^\top \Lambda d \\ -2T^\top \Lambda d \end{bmatrix}, \quad \text{const} = d^\top \Lambda d,$$

with $\Lambda = \operatorname{diag}(\lambda) \in \mathbb{Z}^{m \times m}$.

## 4   Implementation Details

QUBO reformulation of QIP was implemented using an open-source package, `ToQUBO.jl`[3]. This package allows for conversion of JuMP models of QIP into QUBOs and provides an interface with various classical and quantum solvers. Additional details of implementation can be found in reference [14].

A similar approach was taken for QUIO reformulation. Another open-source package, `ToQUIO.jl`[4] was developed for conversion to QUIO models. This tool also builds on the interface provided by the JuMP [4] algebraic modeling language and its low-level backend MathOptInterface (MOI) [3]. Users define a QIP in JuMP as usual, but are able to wrap their QUIO-capable solver constructor within a `ToQUIO.Optimizer` instance, which will automatically construct the QUIO form and handle the necessary transformations regarding model data and results.

More specifically, The translation procedure reads the constraint structure from MOI, adds slack variables, and augments the objective with a penalty term. Penalty weights $\mu \in \mathbb{R}_+^m$ are introduced in a diagonal matrix $M = \operatorname{diag}(\mu)$, which penalizes constraint violations with the quadratic form $rMr$ that follows from the residual $r = Ax + s - b$. Users may supply $\mu$ manually or rely on a default heuristic that sets all $\mu_i = \delta_+ - \delta_-$, where

---

[3] `https://github.com/JuliaQUBO/ToQUBO.jl`
[4] `https://github.com/SECQUOIA/ToQUIO.jl`

$$\delta_+ = \sum_i \max(g_i u_i, g_i \ell_i) + \sum_{i,j} \max(G_{i,j} \ell_i \ell_j, G_{i,j} \ell_i u_j, G_{i,j} u_i \ell_j, G_{i,j} u_i u_j)$$

$$\delta_- = \sum_i \min(g_i u_i, g_i \ell_i) + \sum_{i,j} \min(G_{i,j} \ell_i \ell_j, G_{i,j} \ell_i u_j, G_{i,j} u_i \ell_j, G_{i,j} u_i u_j)$$

This guarantees that any variation in the objective function that happens within variable bounds $[\ell_i, u_i] \ni x_i$ will not be enough to compensate for the penalty incurred upon violation.

Once penalty terms are applied, the package computes the full augmented objective in matrix-vector form $z^\top G z + g^\top z + \text{const}$ as specified in Section 3. Intermediate structures—including variable maps and reformulation metadata—are exposed via standard JuMP interfaces. The package supports solver-agnostic workflows and integrates with `ToQUBO.jl` for side-by-side benchmarking.

We also provide integration with quantum-backend submission via `QCIOpt.jl`[5], a Julia wrapper for the QCI REST API and `QUBODrivers`. This package allows JuMP-based model definitions to interface directly with Quantum Computing Inc. devices. Users can specify attributes such as solver type, device, number of samples via `MOI.set` or the high-level call `JuMP.set_attribute`. Model results—including objective values and samples—are returned in JuMP-compatible structures for analysis and benchmarking. This integration supports both QUBO and QUIO workflows and is used in hybrid quantum-classical experiments.

## 5  Methodology

### 5.1  Test Problems

To evaluate the performance of our QUIO and QUBO encodings, we have defined four optimization problems that use integer decision variables in their QIP form. In the following, we will introduce the these problems that will be used in our evaluation reported in Section 6:

- Quadratic Facility Location Problem (QFLP)
- Quadratic Inventory Management Problem (QIMP)
- Quadratic Vehicle Routing Problem (QVRP)
- Quadratic Knapsack Problem (QKP)

The experiments were carried out on a system running Linux Ubuntu 22.04 with Python 3.10.12 and Julia 1.11. The machine was equipped with an Intel® i7-1365U processor (base frequency 1.80 GHz) and 32.0 GB of memory. The software stack included JuMP v1.26.0, `ToQUBO.jl` v0.1.10, `ToQUTO.jl` v0.1.0, and `QCIOpt` v0.1.0. For solving, we used `Gurobi` v12.0.2 as the branch-and-bound solver and the `Dirac-3` device from Entropy as the quantum backend.

**Quadratic Facility Location Problem**  The Quadratic Facility Location Problem (QFLP) is a complex extension of the classical Facility Location Problem (FLP), where the objective of the latter is to determine optimal locations for facilities at minimal cost of opening them and serving a set of clients. Unlike the linear version, QFLP incorporates quadratic cost terms that account for interactions between pairs of facilities. These interactions might represent synergies or conflicts, such

---

[5] `https://github.com/SECQUOIA/QCIOpt.jl`

as economies of scale, competition, or shared infrastructure costs. The inclusion of these quadratic terms makes the problem non-linear and non-convex, significantly increasing its computational complexity.

QFLP is particularly useful in scenarios where the placement of one facility affects the performance or cost of another. For example, in telecommunications, placing two data centers too close might lead to redundancy, while putting them strategically apart could improve network resilience. Similarly, in retail, opening stores too close to each other might cannibalize sales. Its applications span logistics, urban planning, and network design, making it a vital problem in operations research and strategic decision-making.

Our version of this problem includes the penalization of connectivity between two routes $x$ from $i$ to $j$ and from $k$ to $l$. Additionally, a cost $f$ of opening a number of units $y$ at a facility $j$ is included in the objective function.

$$\text{minimize} \quad \sum_{j=1}^{m} f_j y_j + \sum_{i=1}^{n}\sum_{j=1}^{m}\sum_{k=1}^{n}\sum_{l=1}^{m} c_{ij,kl} x_{ij} x_{kl} \tag{7}$$

$$\text{subject to:} \quad \sum_{j=1}^{m} x_{ij} = d_i \quad \text{for } i = \{1, \ldots, n\} \tag{8}$$

$$\sum_{i=1}^{n} x_{ij} \le M_j y_j \quad \text{for } j = \{1, \ldots, m\} \tag{9}$$

$$x_{ij} \le d_i y_j \quad \text{for } i = \{1, \ldots, n\}; j = \{1, \ldots, m\} \tag{10}$$

$$x_{ij} \ge 0 \quad \text{for } i = \{1, \ldots, n\}; j = \{1, \ldots, m\} \tag{11}$$

$$y_j \le y_{max} \quad \text{for } j = \{1, \ldots, m\} \tag{12}$$

$$y_j \in \mathbb{N} \quad \text{for } j = \{1, \ldots, m\} \tag{13}$$

A demand constraint ensures that all the transactions to one location must match the demand $d$ of this location $i$ as shown in Constraints (8). Constraints (9) limit the sum of transactions that go to one location, restricted by the capacity $M$ of this facility $j$ and the number of facilities opened at the same location $y$. The remaining constraints serve to bound each variable domain.

**Quadratic Inventory Management Problem** The Quadratic Inventory Management Problem (QIMP) is a specialized form of inventory optimization that incorporates a cost function with quadratic terms, reflecting more realistic and complex cost behaviors. Unlike traditional linear models, which assume constant marginal costs, quadratic models account for nonlinear cost structures such as increasing holding costs, penalties for overstocking, or economies of scale in ordering. In our case, we are modeling a setting where we have linear costs $O$ associated with $x$ goods of type $i$ and $y$ goods of type $i$ that we sell. Our quadratic costs in the objective function represent a mixed cost coming from the interaction of two items of different types ($i$ and $j$). The revenue from the sale of $y$ has a unitary cost of $R$. We used the following mathematical formulation for this problem.

$$\text{minimize} \qquad \sum_{i=1}^{n}\sum_{j=1}^{n} H_{i,j} x_i x_j + \sum_{i=1}^{n} O_i x_i - \sum_{i=1}^{n} R_i y_i \qquad (14)$$

$$\text{subject to:} \qquad \sum_{i=1}^{n} x_i \leq s \qquad (15)$$

$$\sum_{i=1}^{n} y_i \leq D \qquad (16)$$

$$y_i - x_i \leq 0 \quad \text{for } i = \{1, \ldots, n\} \qquad (17)$$

$$x_i \in \{0, \ldots, s\} \quad \text{for } i = \{1, \ldots, n\} \qquad (18)$$

$$y_i \in \{0, \ldots, \min(D, s)\} \quad \text{for } i = \{1, \ldots, n\} \qquad (19)$$

where constraint (15) limits the total supply capacity, while Constraint (16) establishes a maximum demand. Constraint (17) states that for each item type $i$, the number of sold items cannot exceed the number of stored items. Finally, the remaining constraints define the integer domains of the decision variables.

**Quadratic Vehicle Routing Problem** The Quadratic Vehicle Routing Problem (QVRP) is an extension of the canonical Vehicle Routing Problem (VRP), where the objective of the former is not only to minimize the total distance or cost of routes but also to account for interactions between pairs of customers or deliveries. In QVRP, the cost function includes quadratic terms that represent these pairwise interactions, which can model real-world complexities such as traffic congestion, delivery synchronization, or environmental impact. This makes the problem significantly more challenging to solve, as the solution space becomes non-linear and more computationally intensive to explore. The problem's complexity and practical relevance make it a rich area for research and application in modern supply chain and urban mobility planning. The QIP formulation we chose for this problem is shown as follows:

$$\text{minimize} \qquad \sum_{i=1}^{n}\sum_{j=1}^{n} D_{i,j} x_i x_j + \sum_{i=1}^{n} T_i x_i \qquad (20)$$

$$\text{subject to:} \qquad \sum_{j=1}^{m} y_j = \sum_{i=1}^{n} x_i \qquad (21)$$

$$x_i \in \{1, \ldots, m\} \quad \text{for } i = \{1, \ldots, n\} \qquad (22)$$

$$y_j \in \{1, \ldots, n\} \quad \text{for } j = \{1, \ldots, m\} \qquad (23)$$

Where the objective function (20) penalizes interactions between pairs of routes $(i,j)$ in relation to the number of vehicles assigned to each of them $x$, considering a cost $D_{i,j}$. Additionally, the base cost of assigning vehicles to route $i$ under a route cost $T_i$ per path $i$ is also computed in the objective function. Constraint (21) was created to affect the number of decision variables in the resulting reformulation, and it can be interpreted as a way of reducing the feasible search space by equalizing the number of vehicles assigned to every route (left-hand side) and the number of vehicles assigned to the routes (right-hand side). Finally, constraints (22) and (23) refer to the decision variables defined on an integer domain.

**Quadratic Knapsack Problem** The Quadratic Knapsack Problem (QKP) consists of selecting $x_i$ items of each kind $i$ so as to maximize the overall value $x^\top Q x$, given that the overall weight of the selected items, i.e. $\sum_i w_i x_i$, does not surpass the maximum capacity $C$. For simplicity, in this example we analyze a variant that is usually known as the *Constrained Quadratic Knapsack Problem*, in which all variables have a pre-determined upper bound $\leq m$. The QIP formulation we chose for this problem is shown as follows:

$$\text{maximize} \qquad \sum_{i=1}^{n} \sum_{j=1}^{n} Q_{i,j} x_i x_j \tag{24}$$

$$\text{subject to:} \qquad \sum_{i=1}^{n} w_i x_i \leq C \quad \text{for } i = 1, \cdots, n \tag{25}$$

$$x_i \in \mathbb{Z}, \quad 0 \leq x_i \leq m \quad \forall i = 1, \ldots, n \tag{26}$$

where the objective function (24) maximizes the total quadratic value between pairs of product types $(i,j)$ with respect to the number of products of each type $x$, considering a cost $Q_{i,j}$. Constraint (25) limits the total weight capacity, and Constraint (26) specifies the domain and bounds of the decision variables, including the range for the number of products of each type $x$.

## 5.2   Hardware implementation

`Dirac-3` is an advanced piece of hardware designed and manufactured by Quantum Computing Inc. (QCI) to solve polynomial optimization problems over integer or continuous variables [8]. Building on a recent research trend within the quantum and physics-inspired hardware community, QCI's machine utilizes quantum noise and decoherence as a fundamental component of its approach, rather than attempting to suppress these phenomena. More importantly, `Dirac-3` implements the *qudit* abstraction [13], a representation of quantum data that supports direct operations over integer numbers instead of binary values as in the usual *qubit* framework.

The vast majority of emergent devices developed for optimization purposes, including `Dirac-3`'s predecessor, Dirac-1, were built around the formulations QUBO and Ising [7]. `Dirac-3` stands out for its native support of integer variables and higher-order polynomials without the need for additional *encoding* and *quadratization* steps. Moreover, besides enjoying full connectivity between its variables with respect to their interactions as nonlinear terms, the machine's capacity is not directly determined by the number of variables it has. Instead, the main bottleneck for representing larger problems lies in its nominal limit on the total number of *levels*, i.e., distinct integer values it can represent across all variables. For example, each binary variable incurs a cost of two levels for the device.

A similar limitation is present when solving problems with continuous variables. In this case, the input model must contain a constraint of the form

$$\sum_{i=1}^{n} x_i = R$$

over every one of its $n$ variables $x_1, \ldots, x_n$ given some value $R$ up to 10000. The resolution for each variable is approximately $R$ divided by the device's *dynamic range*, currently estimated to be approximately 200 [10].

## 6    Computational Experiments

This section includes two kinds of computational experiments. We computed the QUIO and QUBO reformulations of the quadratic integer programs presented above and compared these reformulations. The main object of comparison is the quadratic matrix arising from each reformulation, where we computed its size, equivalent to the number of individual variables to be included in the problem, and its density, accounting for the number of pairwise interaction terms in the quadratic unconstrained program. The second comparison involves solving several instances of one problem with both reformulations using the QCI `Dirac-3` device and comparing the solution to the global optimal solution found using the state-of-the-art branch-and-bound solver `Gurobi`.

### 6.1    Formulation comparisons

We evaluate the QUIO and QUBO formulations on four representative QIP classes: the quadratic facility location problem (QFLP), the quadratic inventory management problem (QIMP), the quadratic vehicle routing problem (QVRP), and the quadratic knapsack problem (QKP). Each case is parameterized by problem size, and we track two key metrics: the number of variables and the density of quadratic terms. All experiments use identical input data, and constraint bounds are scaled to ensure feasibility.

In this evaluation, each case is modeled using a recent version of QUBO.jl [14], which extends the standard implementation that reformulates QIPs to QUBOs (`ToQUBO.jl`) by also supporting reformulation to QUIOs (`ToQUIO.jl`). For all QIPs, their results are reported as follows.

**Quadratid Facility Location Problem (Figure 1, left)**  As the number of clients $n$ increases from 10 to 50, the number of QUBO variables grows rapidly due to binary encoding, exceeding 10,000 in the largest case. In contrast, QUIO exhibits slower scaling and stays under 1,000 variables. The QUBO density increases beyond 0.15 due to the penalty structure and binary slack transformations. QUIO maintains densities below 0.05 across all sizes, reflecting the sparsity preserved from the original constraint matrix $A$.

**Quadratic Inventory Management Problem (Figure 1, right)**  The same trend persists for the knapsack-like inventory management problem. For item counts up to $n = 200$, the QUBO formulation reaches nearly 6,000 variables, whereas QUIO remains compact. Notably, QUBO densities exceed 0.25, a consequence of the dense penalty matrices introduced by binary slack variables. QUIO densities stay under 0.1.

**Quadratic Vehicle Routing Problem (Figure 2, left)**  This class shows a notable saturation behavior. Both QUBO and QUIO densities approach 1.0 as the number of routes grows from 50 to 200. This is because all possible pairwise interactions are activated in the quadratic model, leading to a fully dense structure. However, QUIO still preserves a significant advantage in variable count, remaining under 500 variables, compared to nearly 3,000 variables in QUBO.

**Quadratic Knapsack Problem (Figure 2, right)**  Similar to QVRP, both QUBO and QUIO densities remain at 1.0 across all experiments, regardless of the number of products $n$ or upper

bound $m$. In variable count, QUIO remains under 30 variables, even as $n$ and $m$ increase, reflecting its compact encoding. In contrast, QUBO grows substantially, reaching over 110 variables for the largest problem instances. The gap between QUBO and QUIO becomes more pronounced as either the number of product types or their bounds increases, highlighting the scalability advantage of QUIO.

Across all cases, QUIO consistently yields smaller and sparser formulations. This structural simplicity leads to better numerical conditioning and potentially better scalability with solvers. Empirical trends validate the theoretical motivations introduced in Section 3.1.

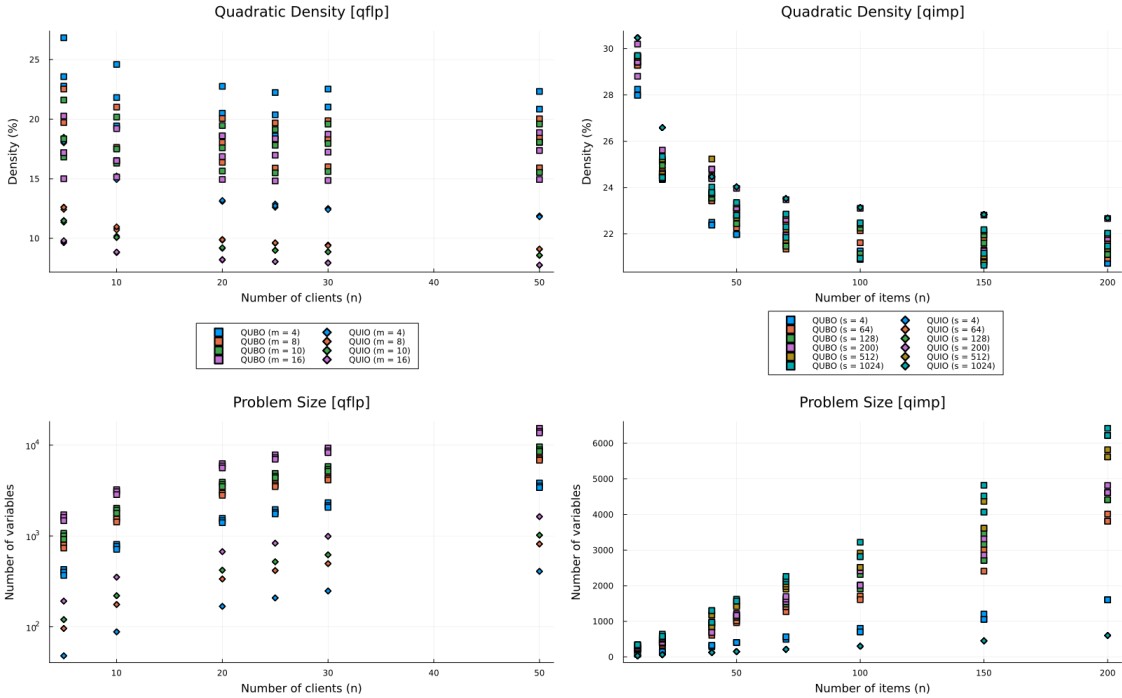

Fig. 1: Variable count and quadratic density vs. number of clients for Quadratic Unconstrained Integer Optimization (QUIO) and Quadratic Unconstrained Binary Optimization (QUBO) reformulations. Quadratic Facility Location Problem (QFLP) (left): QUIO maintains a compact and sparse structure. Quadratic Inventory Management Problem (QIMP) (right): Scaling trends for variable count and density with number of items. QUBO introduces high overhead.

In comparing the QUIO formulation (4) and the QUBO formulation (6), we observe the following empirical implications. QUIO retains $n + m$ integer variables, while QUBO requires $N + K$ binary variables, where $N \gg n$ and $K \gg m$ when $U$ and $V$ are moderate to large. QUIO preserves sparsity from the original constraint matrix $A$, while QUBO introduces denser matrices due to binary encoding and slack transformations. This results in a denser $H$ compared to $G$.

The binary encoding in QUBO often results in poorly conditioned matrices $H$, particularly when the encoding matrix $E$ or slack matrix $T$ contains large coefficients. This can degrade optimization

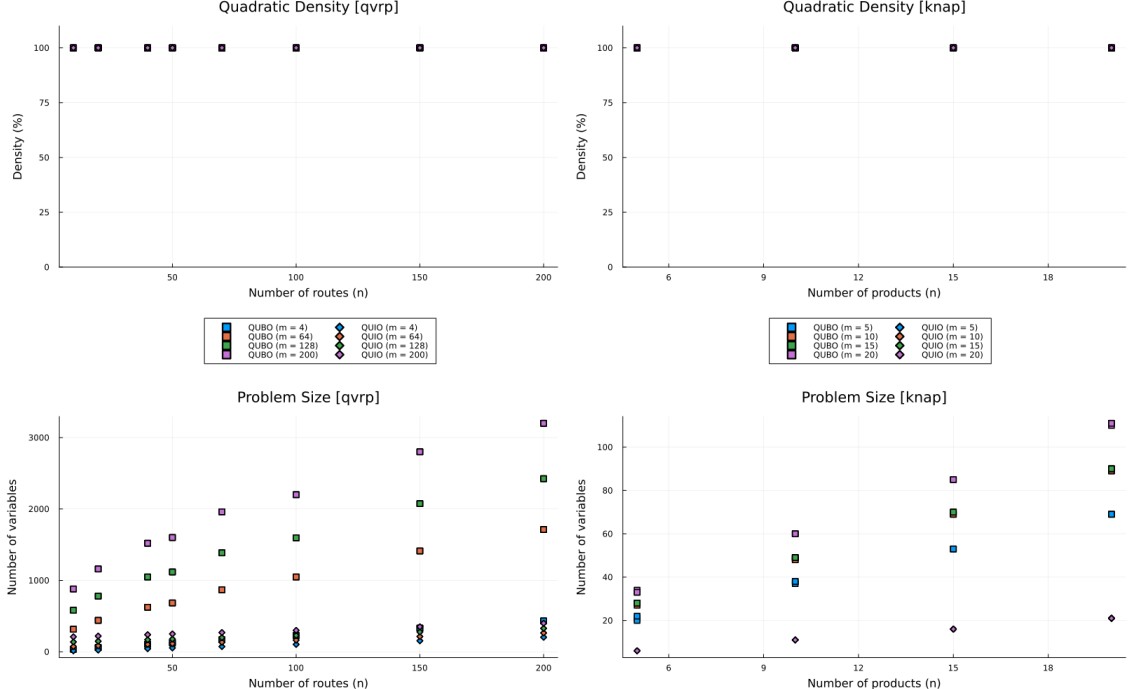

Fig. 2: Variable count and quadratic density vs. number of clients for Quadratic Unconstrained Integer Optimization (QUIO) and Quadratic Unconstrained Binary Optimization (QUBO) reformulations. Quadratic Vehicle Routing Problem (QVRP) (left): Both encodings yield dense matrices, but QUIO remains significantly more compact. Quadratic Knapsack Problem (QKP) (right): Similar to QVRP, the quadratic matrices remain dense across the experiments, and QUBO has faster growth in the number of variables with $n$ than QUIO.

performance. Due to fewer variables and better sparsity, QUIO could be more amenable to classical and quantum heuristics tailored for structured integer spaces, while QUBO suffers from a variable increase.

## 6.2 Hardware Implementation Results

Table 3: Raw Execution Times ($\tau$) and Time to Feasibility ($TTFeas_{99}$) [s] Across 3 Runs for Each $(n, m)$ Instance and Solver

| $n$ | $m$ | Gurobi QIP $\tau$ [s] | Dirac-3 QUBO $\tau$ [s] | Dirac-3 QUBO TTFeas$_{99}$ [s] | Dirac-3 QUIO $\tau$ [s] | Dirac-3 QUIO TTFeas$_{99}$ [s] |
|---|---|---|---|---|---|---|
| 5 | 5 | {0.02, 0.02, 0.02} | {81.94, 79.58, 56.13} | {544.37, 79.58, 214.72} | {77.31, 69.78, 266.42} | {295.71, 69.78, 266.42} |
| | 10 | {0.03, 0.01, 0.06} | {90.21, 93.47, 80.03} | {345.06, 469.79, 306.12} | {73.62, 50.78, 30.17} | {370.02, 337.36, 200.46} |
| | 15 | {0.02, 0.01, 0.09} | {91.56, 88.35, 84.88} | {350.23, 252.79, 426.58} | {82.62, 86.07, 76.80} | {415.22, 86.07, 385.96} |
| | 20 | {0.19, 0.18, 0.07} | {87.87, 95.87, 91.38} | {792.20, 274.32, 261.46} | {75.81, 75.92, 81.81} | {381.00, 290.39, 234.10} |
| 10 | 5 | {0.06, 0.09, 0.02} | {100.36, 96.85, 99.11} | {383.88, 486.76, 379.09} | {75.84, 43.20, 30.11} | {151.67, 86.40, 60.23} |
| | 10 | {0.08, 0.08, 0.46} | {67.03, 68.17, 74.66} | {67.03, 136.34, 285.58} | {35.55, 38.82, 27.97} | {35.55, 38.82, 106.98} |
| | 15 | {0.12, 0.23, 0.36} | {65.93, 116.92, 114.51} | {188.64, 587.61, 229.02} | N/A | N/A |
| | 20 | {0.38, 0.58, 0.41} | {122.12, 133.62, 135.16} | {244.25, 887.76, 516.98} | N/A | N/A |
| 15 | 5 | {0.14, 0.08, 0.07} | {124.00, 122.95, 125.45} | {248.00, 245.91, 479.86} | {55.90, 64.71, 66.17} | {111.81, 129.41, 66.17} |
| | 10 | {0.12, 3.20, 1.72} | {88.70, 144.51, 83.40} | {177.40, 726.30, 238.64} | N/A | N/A |
| | 15 | {0.60, 300+, 0.78} | {153.94, 132.72, 130.85} | {440.49, 265.44, 130.85} | N/A | N/A |
| | 20 | {8.12, 3.05, 300+} | {139.14, 142.68, 131.20} | {398.13, 142.68, 375.40} | N/A | N/A |
| 20 | 5 | {1.27, 3.15, 1.90} | {117.72, 118.12, 131.95} | {235.45, 236.23, 504.71} | {396.15, 45.33, 58.37} | {396.15, 45.33, 58.37} |
| | 10 | {0.90, 0.23, 26.13} | {138.25, 111.75, 130.18} | {276.49, 111.75, 654.29} | N/A | N/A |
| | 15 | {300+, 300+, 105.61} | {148.31, 149.08, 150.28} | {424.37, 570.24, 574.83} | N/A | N/A |
| | 20 | 300+ | {167.05, 189.96, 189.43} | {334.10, 543.54, 378.86} | N/A | N/A |

The QKP example was run on QCI `Dirac-3` to demonstrate the hardware implementation for QUBO and QUIO reformulations. The problem was also modeled using `JuMP` and solved with `Gurobi` for comparison with the QUBO and QUIO reformulations that were automatically generated by `ToQUBO.jl` and `ToQUIO.jl`. For computational time comparison, the time-to-feasibility (TTFeas) was calculated for the `Dirac-3` results to account for their probabilistic nature.

The TTFeas metric is calculated as follows: $TTFeas_s = \tau \cdot \frac{\log(1-s)}{\log(1-p_{feas})}$ where $TTfeas_s$ is the time required to achieve success with probability $s$ (typically 0.99) in reaching feasibility. $\tau$ is the execution time of the algorithm, and $p_{feas}$ is the probability of reaching feasible solutions. If $p_{target} = 1$, which applies for deterministic solvers like `Gurobi`, then $TTFeas_s = \tau$ [6]. The computation execution and calculated $TTFeas_{99}$ times for QKP instances are summarized in Table **??**.

## 7 Discussion and Conclusions

Reformulating and running QIP problems as both QUBO and QUIO reformulated models demonstrated that some of the metrics desirable for optimizing QUBO reformulation are not the same for creating QUIO models. Moreover, the development of (quantum) solvers that natively address QUIO problems, such as `Dirac-3`, motivates the work presented herein. At the same time, it also

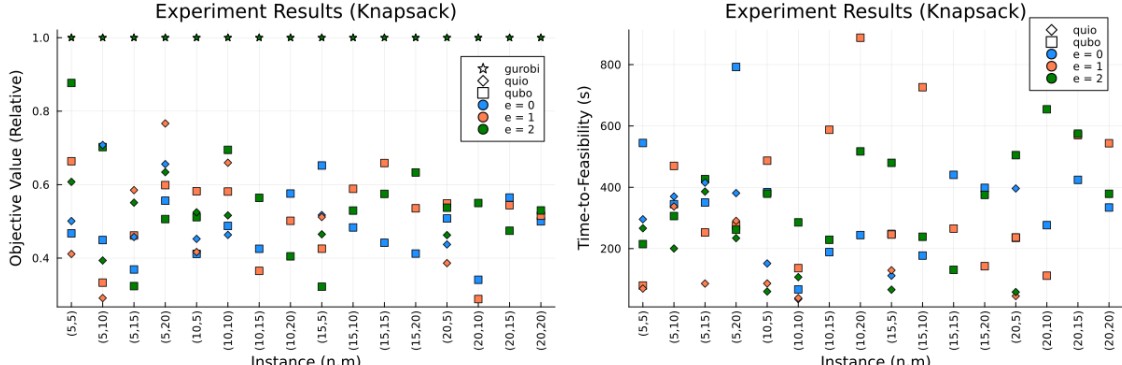

Fig. 3: QKP: (left) hardware run summary-relative objective function value. (right) hardware run summary-ttfeas

presents new challenges to overcome when addressing practical problems in novel hardware. Folklore rules, such as the preference for reformulations that use fewer variables, might be replaced by more elaborate criteria, such as variable levels, whose sum is constrained in `Dirac-3`. This means that instead of trying to use fewer integer variables directly, one should consider using an encoding scheme, as in QUBO, and limit the bounds of those variables accordingly.

On the other hand, using integer variables with more levels brings in some desirable properties, as the produced QUIO (Eq. (4)) models maintain their sparsity in contrast to purely QUBO (Eq. (6)) models, which tend to expand in dimensionality and typically result in a denser matrix of coefficients. Unlike QUBO, the QUIO literature did not evolve significantly towards the development of variable encoding schemes, in part because it was assumed that variables would require no more than translation and scaling transformations to represent decision values. In practice, however, the limitation imposed by the number of levels passed on to the hardware makes the adoption of encoding schemes essential if one wants to explore the hardware's capabilities further. The impact of variable encoding is highlighted by the fact that, despite requiring significantly more variables, QUBO reformulations can fit into the machine more efficiently than their respective QUIO counterparts.

Naturally, these results suggest the need for a deeper understanding of integer variable encoding methods for QUIO. Departing from the binary case, where only the encoding expression is to be decided, one must also choose the appropriate intervals for integer variables. This is undoubtedly a research track in its own right, given the numerous possibilities that can be evaluated as a combination of these two factors. Still within the realm of machine precision, another step forward is to investigate the role that the magnitude of the penalty factors and the overall coefficients of the reformulated problem play in the solver's performance. It also remains unclear whether better heuristics could be adopted to select the penalization factors for QUIO reformulation automatically.

As a final note, we highlight that this is the largest published benchmark of QUIO and QUBO reformulations of quadratic integer programs, only possible through the software development of reformulation code, `ToQUBO.jl` and `ToQUIO.jl`, and integrated solvers with QCI devices via `QCIOpt.jl`. We anticipate these results to spark the curiosity of other researchers into evaluating this innovative hardware for optimization and to develop further algorithms to tackle practical optimization problems using novel computational platforms and problem reformulations.

**Acknowledgments.** The Quantum Collaborative, led by Arizona State University, provided valuable expertise and resources for this project. The Quantum Collaborative connects top scientific programs, initiatives, and facilities with prominent industry partners to advance the science and engineering of quantum information science. This material is based upon work supported by the Center for Quantum Technologies under the Industry-University Cooperative Research Center Program at the US National Science Foundation under Grant No. 2224960. We acknowledge the support of QCI Inc. in providing access to the `Dirac-3` Entropy Quantum Computer and to Wesley Dyk for his invaluable support throughout the development of this paper.

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
