# OpenReview forum: "A Performance Comparison of Variable Encoding Techniques for QUIO and QUBO Problems"
_purdue.edu/Purdue_University/PQAI/2025/Symposium — PQAI 2025 Oral_

### Official Review · Reviewer_82ye · 2025-07-25
**Let's go to higher dimensions, from qubits to qudits**

**Rating:** 7
**Confidence:** 5

**Review:**

This paper compares QUIO and QUBO formulations for solving quadratic integer programs. The focus is on their performance on classical and emerging qudit-based quantum hardware. Variable encoding strategies for QUBO are well-known, but the comparison between QUBO and QUIO in the context of qudit-native devices is relatively new and useful. The paper is well-structured, it has clear formulations, and it includes benchmarks using a real quantum-inspired machine. The software is contributed.

---

### Official Review · Reviewer_znFA · 2025-07-25
**Comparison of formulation for optimisation problems**

**Rating:** 8
**Confidence:** 3

**Review:**

The manuscript “A performance comparison of variable encoding techniques for QUIO and QUBO problems” by Xavier et al. deals with the comparison of performance between two different formulations of the Quadratic Integer Optimisation problems. One of them is the well-known QUBO formulation, whereas the Quadratic Unconstrained Integer Optimization (QUIO) formulation is amenable for the usage on qudit-based systems. The authors select four example problems and compare for them the two different formulations, according to different metrics.

The work is well-structured and presents an interesting research question, in view of novel computing paradigms. I have nevertheless a few concerns about it:

P. 7: the names Quantum Computing Inc. and Dirac-3 are introduced here abruptly without proper references (which actually come later in the text). I suggest that the authors improve the way the system is presented, as some of the readers may not be familiar with it;

Section 5: what is the rationale behind the four chosen use cases? In which way are they representative of the generality of Quadratic Integer Optimisation problems?

Section 6.1: for the first two problems there is a mismatch between the densities described in the text and the numbers I read from the plots;

Section 6.3: besides comparing the time diagnostics, are authors checking that the two formulations are verified against each other, in terms of providing the same results?

---

### Decision · Program_Chairs · 2025-07-29

Accept (Oral)